# Vesicular Glutamate Transporter 3 Is Involved in Glutamatergic Signalling in Podocytes

**DOI:** 10.3390/ijms26062485

**Published:** 2025-03-11

**Authors:** Naoko Nishii, Tomoko Kawai, Hiroki Yasuoka, Tadashi Abe, Nanami Tatsumi, Yuika Harada, Takaaki Miyaji, Shunai Li, Moemi Tsukano, Masami Watanabe, Daisuke Ogawa, Jun Wada, Kohji Takei, Hiroshi Yamada

**Affiliations:** 1Department of Nephrology, Rheumatology, Endocrinology and Metabolism, Okayama University Graduate School of Medicine, Dentistry and Pharmaceutical Sciences, 2-5-1 Shikata-cho, Okayama 700-8558, Japandaiogawa2000@gmail.com (D.O.); junwada@okayama-u.ac.jp (J.W.); 2Department of Cell Physiology, Okayama University Graduate School of Medicine, Dentistry and Pharmaceutical Sciences, 2-5-1 Shikata-cho, Okayama 700-8558, Japan; tokawai@okayama-u.ac.jp; 3Department of Neuroscience, Okayama University Graduate School of Medicine, Dentistry and Pharmaceutical Sciences, 2-5-1 Shikata-cho, Okayama 700-8558, Japantabe@md.okayama-u.ac.jp (T.A.); kohji@md.okayama-u.ac.jp (K.T.); 4Department of Genomics and Proteomics, Advanced Science Research Center, Okayama University, Okayama 700-8530, Japan; y-harada@okayama-u.ac.jp (Y.H.); miyaji-t@okayama-u.ac.jp (T.M.); 5Center for Innovative Clinical Medicine, Okayama University Hospital, Okayama 700-8558, Japan; lishunai@md.okayama-u.ac.jp (S.L.); masami5@md.okayama-u.ac.jp (M.W.); 6Central Research Laboratory, Okayama University Graduate School of Medicine, Dentistry and Pharmaceutical Sciences, 2-5-1 Shikata-cho, Okayama 700-8558, Japan

**Keywords:** VGLUT3, glutamate, podocyte, glutamatergic transmission

## Abstract

Glomerular podocytes act as a part of the filtration barrier in the kidney. The activity of this filter is regulated by ionotropic and metabotropic glutamate receptors. Adjacent podocytes can potentially release glutamate into the intercellular space; however, little is known about how podocytes release glutamate. Here, we demonstrated vesicular glutamate transporter 3 (VGLUT3)-dependent glutamate release from podocytes. Immunofluorescence analysis revealed that rat glomerular podocytes and an immortal mouse podocyte cell line (MPC) express VGLUT1 and VGLUT3. Consistent with this finding, quantitative RT-PCR revealed the expression of VGLUT1 and VGLUT3 mRNA in undifferentiated and differentiated MPCs. In addition, the exocytotic proteins vesicle-associated membrane protein 2, synapsin 1, and synaptophysin 1 were present in punctate patterns and colocalized with VGLUT3 in MPCs. Interestingly, approximately 30% of VGLUT3 colocalized with VGLUT1. By immunoelectron microscopy, VGLUT3 was often observed around clear vesicle-like structures in differentiated MPCs. Differentiated MPCs released glutamate following depolarization with high potassium levels and after stimulation with the muscarinic agonist pilocarpine. The depletion of VGLUT3 in MPCs by RNA interference reduced depolarization-dependent glutamate release. These results strongly suggest that VGLUT3 is involved in glutamatergic signalling in podocytes and may be a new drug target for various kidney diseases.

## 1. Introduction

Glomerular podocytes are highly differentiated epithelial cells that line the urinary side of the glomerular basement membrane of the kidney and play a key role in blood filtration. Podocytes on capillaries form complex processes that interdigitate with those of neighbouring podocytes to form and maintain glomerular slit diaphragms [1].

Glutamate is a major intercellular transmitter in the nervous system that activates ionotropic and metabotropic glutamate receptors [2]. Endocrine tissues, such as the pineal gland and pancreatic islets, also use glutamate to transmit signals [3,4]. Glomerular podocytes express functional ionotropic NMDA receptors, metabotropic glutamate receptor 1 (mGluR1), and mGluR5 receptors [5,6,7,8,9]. Stimulation of these receptors is involved in the formation and maintenance of slit diaphragms and podocyte injury [5,6,7,8,9], indicating that the glutamatergic system in podocytes has physiological importance.

The glutamatergic system consists of the release, reception, and termination of glutamate signals. To transmit glutamate signals, glutamate must be taken up by vesicular-type glutamate transporters (VGLUTs) and stored in synaptic vesicles. There are three VGLUT isoforms, VGLUT1, VGLUT2, and VGLUT3 [10]. All VGLUT isoforms are localized at synaptic vesicle membranes and commonly transport cytoplasmic glutamate into vesicles using the membrane potential coupled with ATP hydrolysis by V-ATPase [10]. VGLUT1 is expressed in podocytes [11], and VGLUT3 is expressed in the kidney, suggesting that the kidney contains cells, including podocytes, which transmit glutamate signals [12].

The ionotropic NMDA receptor in mature podocytes is involved in the progression of diabetic nephropathy [8], and NMDA receptor antagonism in cultured podocytes alters cytoskeletal remodelling and the integrity of the glomerular filtration barrier [13]. Metabotropic mGluR1 and mGluR5 are expressed in the foot processes of podocytes, and their activation improves albumin levels and apoptosis, which are processes that depend at least on cAMP [5]. Thus, accumulating evidence suggests the existence of a glutamate-signalling system in glomerular podocytes. However, little is known about its molecular mechanisms. In this study, we investigated the functional expression of VGLUT3 in differentiated mouse podocyte cell lines (MPCs) and observed that VGLUT3 largely colocalized with VGLUT1 and VAMP2, as well as with the synaptic vesicle marker proteins synaptophysin 1 and synapsin 1. Furthermore, immunoelectron microscopy showed that VGLUT3 was present on the membrane of vesicle-like structures. Differentiated MPCs secreted glutamate into the extracellular space upon stimulation with pilocarpine or depolarization with KCl. The depletion of VGLUT3 in differentiated MPCs decreased the KCl-induced glutamate release. These results suggest that podocytes can release glutamate in a VGLUT3-dependent manner and VGLUT3 is likely important for regulating the glomerular slit diaphragm.

## 2. Results

### 2.1. VGLUT3 Is Present in Glomerular Podocytes

To investigate the expression of VGLUT1 and VGLUT3 in rat podocytes, immunohistochemistry was performed. Consistent with a previous report [11], VGLUT1 immunoreactivity was clearly observed in glomeruli, which localize at the periphery of glomerular capillaries (Figure 1A). VGLUT3 was also observed in glomeruli, and the staining pattern was similar to that of VGLUT1 (Figure 1B). Furthermore, both VGLUT1 and VGLUT3 were colocalized with synaptopodin, a podocyte marker, indicating their presence in glomerular podocytes (Figure 1).

### 2.2. VGLUT1 and VGLUT3 Are Co-Expressed in Differentiated MPCs

Given the presence of VGLUT1 and VGLUT3 proteins in rat podocytes, we next examined their expression in an MPC. MPCs can be differentiated by changing the culture temperature from 33 °C to 37 °C, along with the removal of γ-interferon from the culture medium [14]. First, we examined the mRNA expression of VGLUT1, VGLUT2, and VGLUT3 in MPCs by real-time PCR. Both VGLUT1 and VGLUT3 mRNAs were detected in undifferentiated and differentiated MPCs, whereas VGLUT2 mRNA was undetectable under our experimental conditions. Furthermore, VGLUT3 mRNA was markedly increased during differentiation (Figure 2A). Western blotting analyses detected expression of VGLUT1 and VGLUT3 proteins in homogenates prepared from undifferentiated or differentiated MPCs (Figure 2B and Appendix A). Next, the localization of VGLUT1 or VGLUT3 in MPCs was examined by double immunofluorescence. In undifferentiated and differentiated cells, VGLUT1 and VGLUT3 were present as punctata scattered throughout the cell, and the two proteins significantly colocalized (Figure 2C,D). The colocalization rate of VGLUT3 with VGLUT1 was 30.3 ± 1.0% (*n* = 33 cells) in undifferentiated cells and 37.0 ± 0.62% (*n* = 33 cells) in differentiated cells (Figure 2E).

### 2.3. VGLUT3 Colocalizes with Synaptic Vesicle Marker Proteins in Differentiated MPCs

In the brain, VGLUT3 is localized at synaptic vesicles and is involved in the exocytosis of transmitters [10]. Next, the colocalization of VGLUT3 with vesicular exocytosis-related proteins, namely vesicle-associated membrane protein 2 (VAMP2), synaptophysin 1, and synapsin 1, was determined in differentiated MPCs. As shown in Figure 3A–C, these proteins showed the same punctate pattern as VGLUT3 and were colocalized with VGLUT3. The colocalization rate of VGLUT3 with these proteins was 30–40% (Figure 3D). These results suggest that vesicular trafficking using VGLUT3-containing synaptic-like vesicles takes place in differentiated MPCs.

### 2.4. VGLUT3 Associates with Synaptic-like Vesicles in Differentiated MPCs

Ultrastructural observation of differentiated MPCs by transmission microscopy revealed that podocytes contain a considerable number of small, clear vesicles that are 100–150 nm in diameter (Figure 4A). By immunoelectron microscopy, VGLUT3 was present on the membrane of these vesicles (Figure 4B). Taken together, these results indicate that podocytes express VGLUT3-positive synaptic-like vesicles, which could contain glutamate that is released into the intercellular space by exocytosis upon some kind of stimulus.

### 2.5. VGLUT3 Depletion in Differentiated MPCs Reduces Glutamate Release into the Intracellular Space

Since our results strongly suggested the presence of glutamate transport and release in podocytes, we directly measured glutamate release from differentiated MPCs. The glutamate concentration in the culture medium was determined by HPLC after 5 min stimulation of the cells with pilocarpine or high K^+^. Stimulated podocytes released a considerable amount of glutamate, 0.54 ± 0.17 or 0.94 ± 0.14 µM, following the addition of pilocarpine or high K^+^, respectively (Figure 5A). By contrast, nicotine, an agonist for nicotinic acetylcholine receptors, had little effect on the release. These results are consistent with a previous report [11]. Finally, we examined whether the expression of VGLUT3 is essential for the glutamate release from differentiated MPCs, utilising RNA interference (RNAi) to reduce the expression of VGLUT3. Immunofluorescence revealed that the expression of VGLUT3 was selectively knocked down by approximately 50% compared with control cells (Appendix A). High K^+^-dependent glutamate release was reduced by approximately 60% compared with control MPCs (Figure 5B). These results suggest that VGLUT3 is functionally expressed by MPCs and is involved in glutamate release into the intracellular space.

## 3. Discussion

In the current study, we investigated the expression, intracellular localization, and possible roles of VGLUTs in conditionally immortalised mouse podocyte cell lines (MPCs). We demonstrated the expression of VGLUT1 and VGLUT3 in MPCs as well as renal glomerular podocytes (Figure 1 and Figure 2). In differentiated MPCs, VGLUT3 was colocalized with synaptic vesicle markers VGLUT1, VAMP2, synaptophysin 1, and synapsin 1 (Figure 2). Furthermore, the presence of VGLUT3 on the membrane of intracellular vesicles was indicated by immunoelectron microscopy (Figure 4). These results suggested podocytes can release glutamate into the extracellular space through vesicle-mediated exocytosis. Finally, we demonstrated that MPCs can release glutamate upon depolarization with a high concentration of K^+^ or the muscarinic agonist pilocarpine but not nicotine. RNAi-mediated depletion of VGLUT3 reduced KCl-dependent glutamate release (Figure 5). Taken together, these results suggest that VGLUT3 is functionally expressed in differentiated MPCs and that it might be involved in the glutamatergic regulation of podocyte functions.

VGLUT3 was identified by several groups in 2002 as a third VGLUT isoform [12,15,16,17] and was characterised in the brain and neurons in comparison with VGLUT1 and VGLUT2. In humans, VGLUT3 shares approximately 80% amino acid homology and has almost identical functional properties such as glutamate uptake and substrate specificity as VGLUT1 and VGLUT2, suggesting that VGLUT3 contributes to exocytotic glutamate release. Few studies have investigated VGLUTs in non-neuronal cells. Although VGLUT3 mRNA is present in the liver and kidney as well as the brain [12], the expression of VGLUT3 at a protein level was not known before the current study.

First, Rastaldi and colleagues demonstrated the expression of vesicular exocytosis-related proteins, including rab3A, rabphillin-3A, synaptotagmin 1, synapsin 1, synaptophysin 1, VGLUT1, and glutamate release in podocytes [11,18]. Following these reports, various glutamate receptors and their physiological importance in podocytes were reported [5,6,7,8,9], and the importance of glutamate signalling in regulating podocyte function was established. However, it remained unclear how glutamate was released from podocytes. Glutamate is secreted from non-neuronal cells by exocytosis and non-exocytosis [19]. We showed that glutamate release is largely dependent on VGLUT3 in podocytes. Furthermore, VGLUT3 considerably colocalized with VGLUT1. These findings suggest that at least a subpopulation of VGLUT3 is engaged in the exocytosis of glutamate. Other subpopulations of VGLUT3 might be involved in glutamate storage or an as-yet-unknown “novel mode” of glutamate signalling, which is proposed to occur in neurons [12]. The mRNA expression of VGLUT3 was enhanced in differentiated podocytes as compared to that in undifferentiated podocytes (Figure 2A). Comparatively, the protein expression of VGLUT3 was not increased in differentiated podocytes (Figure 2B). Since the activity of many proteins, including synaptopodin, CD2AP, and dynamin, is regulated by degradation with several proteases in mature podocytes [20], VGLUT3 could be regulated by the degradation.

There may also be a non-exocytotic mechanism of glutamate release from podocytes. Extracellular glutamate is taken up by Na^+^-dependent glutamate transporters on the plasma membrane, which serve as terminators of glutamate signals. These transporters can run in reverse and release glutamate [21]. Excitatory amino acid transporter 2 (EAAT2) is expressed in podocytes [22], and it can reverse and release glutamate. Another possible mechanism involves channels. Astrocytes release glutamate via the P2X7 receptor [23], which has a large pore that is permeable to ATP and glutamate [24]. Podocytes express P2X7 [25]. Although further studies are needed, it is possible that podocytes may be able to secrete glutamate using the VGLUT3-mediated exocytosis discovered in this study and unknown non-exocytotic release.

In this study, pilocarpine, a muscarinic acetylcholine receptor agonist, stimulated glutamate release from cultured differentiated podocytes, which is consistent with a previous report [11]. In addition, depolarising stimuli also triggered glutamate release (Figure 5). Pilocarpine increases intracellular Ca^2+^ concentration [26]. The depolarization stimuli open N-type voltage-dependent calcium channels, which are present in podocytes [27]. Since both pilocarpine and depolarization increased the intracellular calcium concentration, it is conceivable that muscarinic acetylcholine receptors and N-type voltage-dependent calcium channels are involved in VGLUT3-dependent glutamate release. The identification of the physiological stimuli that promote glutamate release is needed in future studies.

Previous reports suggest that glutamate signalling is essential for the morphological regulation of the slit diaphragm formed by the interdigitation of foot processes of neighbouring podocytes and apoptosis [5,8,9]. In mature podocytes, the ionotropic NMDA receptor is involved in the progression of diabetic nephropathy [8]. Furthermore, metabotropic mGluR1 and mGluR5 are functionally expressed in differentiated podocytes. In particular, mGluR1 and mGluR5 are expressed in the foot processes of podocytes, and their activation improves albumin levels and apoptosis, which are processes that depend on cAMP [5]. Thus, glutamate signalling and the signal reception machinery expressed by mature podocytes are crucial for controlling blood filtration function, and VGLUT3 is an essential component of this machinery.

In this study, we found that podocytes themselves release glutamate via the activity of VGLUT3. By adjusting the glutamate concentration by regulating VGLUT3 activity, it would be possible to regulate the input of glutamate signals and the podocyte functions. Thus, the podocyte glutamatergic system including VGLUT3 will likely become important as a drug target for protecting podocytes from injury and improving proteinuria.

## 4. Materials and Methods

### 4.1. Antibodies and Reagents

Guinea pig polyclonal anti-vesicular glutamate transporter 3 (anti-VGLUT3) antibodies (cat# AB5421-I), rabbit polyclonal anti-synapsin 1 antibodies (cat# AB1543P), and rabbit polyclonal anti-glutamate antibodies (cat# AB133) were purchased from Merck Millipore (Darmstadt, Germany). Rabbit polyclonal anti-synaptobrevin 2 (anti-VAMP2) antibodies (cat# 104202), rabbit polyclonal anti-synaptophysin 1 antibodies (cat# 101002), and rabbit polyclonal anti-vesicular glutamate transporter 1 (anti-VGLUT1) antibodies (cat#135303) were purchased from Synaptic Systems GmbH (Göttingen, Germany). A mouse monoclonal antibody against beta-actin (cat# A5441) was purchased from Sigma-Aldrich (St. Louis, MO, USA). A mouse monoclonal antibody against synaptopodin (clone G1D4, cat# 65194) was purchased from PROGEN Biotechnic GmbH (Heidelberg, Germany). Alexa Fluor 488-conjugated donkey anti-rabbit IgG (cat# A21206) or Alexa Fluor 555-conjugated donkey anti-rabbit IgG (cat# A31572), Alexa Fluor 555- conjugated donkey anti-mouse IgG (cat# A31570), and Alexa Fluor 488-conjugated goat anti-mouse IgG antibodies (cat# A11001) were purchased from Thermo Fisher Scientific (Waltham, MA, USA). Alexa Fluor 488-conjugated donkey anti-guinea pig IgG (cat# 706-546-148) and peroxidase-conjugated donkey anti-guinea pig IgG (cat# 706-035-148) antibodies were purchased from Jackson ImmunoResearch (West Baltimore, PA, USA). The 10 nm gold-conjugated secondary antibodies (cat# EMGAG10) were purchased from BBI Solutions (ME, USA). Nicotine (cat#140-01211), pilocarpine hydrochloride (cat# 161-07201), and O-phthalaldehyde (cat# 167-09263) were purchased from Wako-Fujifilm Co., Ltd. (Osaka, Japan). DL-TBOA (Threo-β-Benzyloxyaspartic acid, cat#1223) was purchased from Tocris Bioscience (Bristol, UK).

### 4.2. Cell Culture

The conditionally immortalised mouse podocyte cell line (MPC) was cultured as described previously [14]. Briefly, the cells were cultured on type I collagen-coated plastic dishes (cat# 356450; Corning Inc., Corning, NY, USA) in RPMI 1640 medium (cat# 189-02025; Fujifilm Wako Pure Chemicals Co. Ltd., Tokyo, Japan) containing 10% foetal bovine serum (cat# 10100147, Thermo Fisher Scientific), 100 U/mL penicillin, 100 µg/mL streptomycin (cat# 15140122, Thermo Fisher Scientific), and 50 U/mL mouse recombinant γ-interferon (cat# 315-05; PeproTech, Rocky Hill, NJ, USA), and were maintained at 33 °C and 5% CO_2_. For differentiation, podocytes were cultured at 37 °C in medium lacking γ-interferon for 7–14 days. Under these conditions, the cells stopped proliferating and became positive for synaptopodin expression.

### 4.3. siRNA-Mediated Interference and Transfection

The pre-annealed siRNA mixture for mouse VGLUT3 (cat# 4390771) was purchased from Ambion (Thermo Fisher Scientific), and the negative control (cat# D0018101005) siRNAs were synthesised and purified by Dharmacon Inc. (Lafayette, CO, USA). The siRNAs targeting independent sequences of mouse VGLUT3 were mixed: oligo 1 sense, 5′-GGACAAAUGUGGAAUCAUU-3′. Scrambled RNA with no significant sequence homology to the mouse, rat, or mouse VGLUT3 gene sequence was used as the negative control. Undifferentiated MPCs were transfected with the siRNAs using Lipofectamine RNAiMax reagent (cat# 13778-150, Thermo Fisher Scientific). The cells were seeded either into 6-well plates coated with type I collagen (cat# 356400, Corning Inc.) at a density of 1 × 10^3^ cells/well for immunostaining experiments or a 10 cm diameter culture dish coated with type I collagen (cat #4020-010, AGC Techno Glass Co., Ltd., Shizuoka, Japan) at a density of 1 × 10^4^ cells/dish for glutamate release.

Two days later, each well was incubated for 6 h with 150 pmol siRNA and 7.5 µL RNAiMax in Opti-MEM (cat#31985070, Thermo Fisher Scientific) containing γ-interferon. Each dish was incubated with 600 pmol siRNA and 35 µL RNAiMax in Opti-MEM containing γ-interferon. Subsequently, the transfection medium was replaced with fresh medium containing γ-interferon. After 72 h, a second transfection was performed, and the cells were cultured for another 72 h. It was confirmed that the siRNAs reduced the expression of VGLUT3 by approximately 50% (Appendix A). To enable differentiation, the cells were plated into new culture dishes and maintained in medium lacking γ-interferon at 37 °C for 7 days.

### 4.4. Quantitative PCR

Total RNA was prepared from MPCs using an RNeasy mini kit (Qiagen, Hilden, Germany) according to the manufacturer’s instructions. cDNA was generated using a PrimeScript RT master mix (Takara Bio, Shiga, Japan) with 1 µg of total RNA as the template. Quantitative PCR was carried out with specific forward and reverse primers at 0.25 µM and Luna universal qPCR master mix (New England Biolabs, Ipswich, MA, USA) in a 20 µL reaction volume using a QuantStudio3 PCR system (Thermo Fisher Scientific). Reaction conditions included an initial denaturation step of 95 °C for 60 s, followed by 40 cycles of denaturation at 95 °C for 15 s, and annealing/extension at 60 °C for 30 s. The primer sets used for the detection of mouse *Vglut1* (140 bp), mouse *Vglut2* (142 bp), and mouse *Vglut3* (124 bp) were as follows: 5’-TTGAAGAAGTGTTCGGCTTTGAGA-3’ and 5’-GTTGGTAGTGGACATTATGTGACGA-3’; 5’-AGCAAATTCTCTCAACAACTACAGTG-3’ and 5’-CTGAATCCTACTGCAAGCACCAA-3’; 5’-TTACGGCTGTGTCATGGGTGT-3’ and 5’-AGTCGTGGCTAGACGGCTTC-3’, respectively. The level of *Vglut1-3* was evaluated relative to that of the housekeeping gene, glyceraldehyde 3-phosphate dehydrogenase (*G3pdh*). The primer set used for the detection of mouse *G3pdh* (150 bp) was as follows: 5’-TGTGTCCGTCGTGGATCTG-3’ and 5’-TTGCTGTTGAAGTCGCAGGAG-3’.

### 4.5. Immunohistochemistry

Immunohistochemistry was performed on renal glomerular podocytes obtained from rats. Under sevoflurane anaesthesia, 7-week-old male Wister rats (Shimizu Laboratory Supplies Co., Kyoto, Japan) were perfusion-fixed with 4% paraformaldehyde and 20% sucrose in phosphate-buffered saline (150 mM NaCl, 10 mM phosphate buffer, pH 7.4). The kidney was removed, then cut into sections and fixed with the same fixative at 4 °C for 16 h. The fixed kidney was cryoprotected with 18% sucrose and frozen-sectioned at 8 µm thickness. Dissected kidney obtained from 2-day-old rats under sevoflurane anaesthesia was fixed in the same fixative and was frozen-sectioned at 8 µm thickness. The sections were double-stained by immunofluorescence, as described previously [28].

### 4.6. Electron Microscopy

Pre-embedding immunoelectron microscopy of cultured podocytes was performed as described previously [29]. Briefly, differentiated MPCs were fixed with 4% paraformaldehyde in 0.1 M phosphate buffer, pH 7.4, for 15 min, and then washed once. Cells were permeabilised with 0.25% saponin in 0.1 M phosphate buffer for 30 min. After incubation in blocking solution (1% bovine serum albumin and 10% goat serum in 0.1 M PB) for 15 min, samples were incubated with guinea pig polyclonal anti-VGLUT3 antibody (1:750) diluted in blocking solution at 4 °C for 16 h, washed with 1% bovine serum albumin in 0.1 M phosphate buffer five times, incubated with 10 nm gold-conjugated secondary antibodies (1:50), and then fixed with 1% glutaraldehyde in 0.1 M phosphate buffer for 5 min. The samples were post-fixed with 0.5% osmium tetroxide in 0.1 M sodium cacodylate buffer for 90 min, dehydrated, and embedded in Epon 812 embedding resin (cat# 341; Nissin EM Co., Ltd., Tokyo, Japan) to enable ultra-thin sectioning. The sections were observed with a Hitachi H-7650 transmission electron microscope (Hitachi High-Tech Corp., Tokyo, Japan). For conventional electron microscopy, cell specimens were immersed in 0.1 M cacodylate buffer, pH 7.4, containing 2% glutaraldehyde and 2% paraformaldehyde for 16–18 h. Post-fixation was performed in 2% osmium tetroxide for 1.5 h. After washing with cacodylate buffer, the specimens were dehydrated in a graded series of ethanol solutions and embedded in Spurr resin (Polysciences Inc., Warrington, PA, USA). Then, 80 nm sections were prepared using an ultramicrotome (EM-UC7; Leica Biosystems, Nussloch, Germany) and stained with uranyl acetate and lead citrate. Specimens were observed by transmission electron microscopy (H-7650, Hitachi, Tokyo, Japan).

### 4.7. Fluorescent Microscopy

MPCs were fixed with 4% paraformaldehyde and stained by immunofluorescence, as described previously [28]. VGLUT1, VGLUT3, synapsin 1, synaptophysin 1, VAMP2, and glutamate were visualised by double immunofluorescence. Samples were examined using a spinning disc confocal microscope system (X-Light confocal imager; Crestoptics S.p.A., Rome, Italy) combined with an inverted microscope (IX-71; Olympus Optical Co., Ltd., Tokyo, Japan) and an iXon+ camera (Oxford Instruments, Oxfordshire, UK). The confocal system was controlled by MetaMorph version 7.10.3.279 (Molecular Devices, Sunnyvale, CA, USA). When necessary, images were processed using Adobe Photoshop version 26.4.1 or Illustrator version 29.3.1.

### 4.8. Determination of Glutamate Release

Cultured cells (1 × 10^4^ cells per 10 cm diameter dish) were washed three times with Ringer’s solution comprising 128 mM NaCl, 1.9 mM KCl, 1.2 mM KH_2_PO_4_, 2.4 mM CaCl_2_, 1.3 mM MgSO_4_, 26 mM NaHCO_3_, 3.3 mM glucose, and 10 mM HEPES (pH 7.4). After cells had been incubated in 5 mL of Ringer’s solution at 37 °C, glutamate release was stimulated by the addition of the indicated stimulants or high K^+^ Ringer’s solution (83 mM NaCl, 50 mM KCl, 1.2 mM KH_2_PO_4_, 2.4 mM CaCl_2_, 1.3 mM MgSO_4_, 26 mM NaHCO_3_, 3.3 mM glucose, and 10 mM HEPES (pH 7.4)). All media contained 50 µM DL-TBOA, an inhibitor for Na^+^-dependent glutamate transporter for decreasing glutamate reuptake into podocyte. Aliquots (500 µL) of the incubation media were taken at specific times, and the amount of extracellular glutamate was determined by high-performance liquid chromatography (HPLC) with precolumn O-phthalaldehyde derivatisation, separation on a reverse-phase Resolve C18 column (4.6 × 150 mm) (cat#38144-31; Nacalai Tesque Inc., Kyoto, Japan), and fluorescence detection [30].

### 4.9. Morphometry

To assess the colocalization of VGLUT3 with VGLUT1, VAMP2, synaptophysin 1, or synapsin 1, immunostained cells were imaged, and the immunoreactivities in three randomly selected areas (25 µm^2^ each, total 75 µm^2^) per cell were measured using MetaMorph software. To determine the effect of VGLUT3 knockdown in MPCs, images of control or VGLUT3 siRNA-treated cells stained with an antibody against VGLUT3 were acquired at identical settings with a 60× objective for cells. The average pixel intensities per cell were then measured using Image J version 1.52a. To enable the fluorescence to be quantified, a background correction was performed before each measurement.

### 4.10. Ethics and Animal Use Statement

All experiments and protocols were approved by the institutional animal care and use committee of Okayama University (OKU-2024424, 15 April 2024, Okayama, Japan). All efforts were made to minimise animal suffering. Whole kidneys were removed from mice after euthanising.

### 4.11. Statistical Analysis

Data were analysed for statistical significance using KaleidaGraph software for Macintosh, version 5.0.6 (Synergy Software Inc., Essex Junction, VT, USA). Student’s *t*-tests were used to analyse two groups. *p* < 0.05 was considered significant.

## 5. Conclusions

We have investigated the mode of glutamate release from renal podocytes. Podocytes have vesicular glutamate transporter 3 in addition to VGLUT1. Glutamate release from podocytes is triggered by depolarization with a high concentration of KCl or pilocarpine, suggesting that the glutamate release involves cholinergic signal transmission. Thus, podocytes use glutamate secreted, at least in part, in a VGLUT3-dependent manner to regulate their morphology and function as a filter barrier of blood.

## Figures and Tables

**Figure 1 ijms-26-02485-f001:**
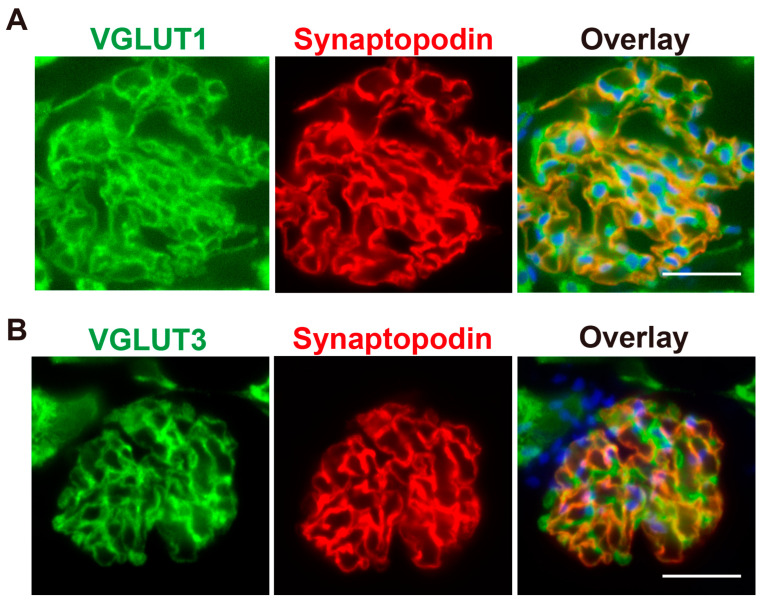
VGLUT1 and VGLUT3 are expressed in podocytes from rat kidney. (**A**) The distribution of VGLUT1 (**left**) in the rat renal glomerulus. Sections were co-stained for synaptopodin (**middle**). The overlay image includes nuclear staining (**right**). Bar: 30 µm. (**B**) The distribution of VGLUT3 (**left**) in the rat renal glomerulus. Sections were co-stained for synaptopodin (**middle**). The overlay image includes nuclear staining (**right**). Bar: 30 µm.

**Figure 2 ijms-26-02485-f002:**
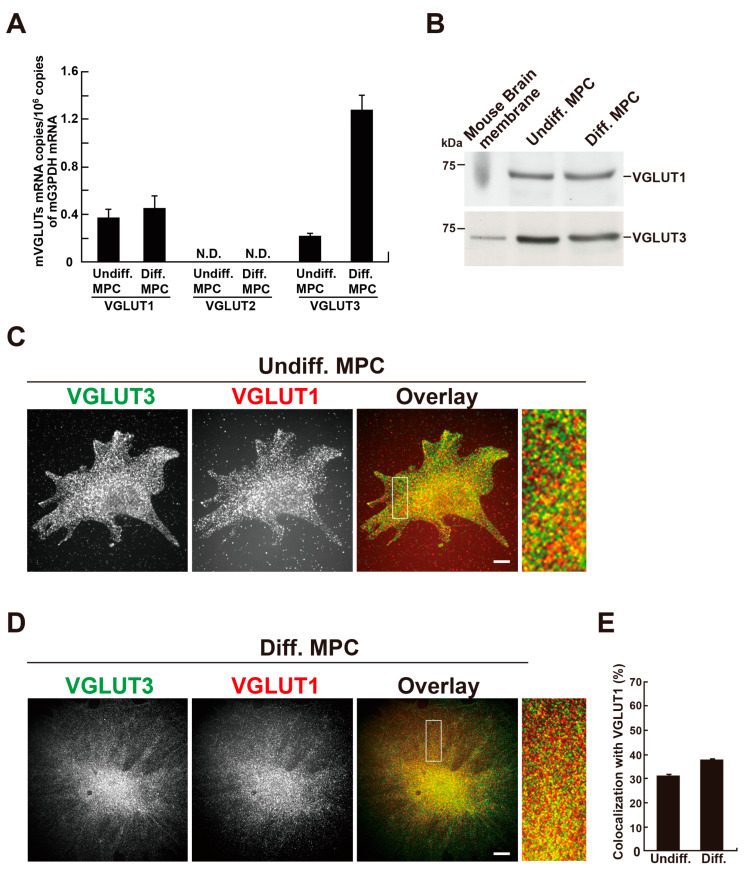
VGLUT3 is expressed and partially colocalizes with VGLUT1 in a differentiated mouse podocyte cell line (MPC). (**A**) *Vglut1-3* gene expression in undifferentiated and differentiated MPCs. The expression was examined by quantitative PCR analysis with total RNA from undifferentiated and differentiated MPCs using probes specific to mouse *Vglut1-3* or *G3pdh*. The levels of *Vglut1-3* mRNA were defined as copies relative to those of *G3pdh*. N.D. means “not detected”. Data are means ± S.E.M., *n* = 7–8. (**B**) Western blot analyses of VGLUT1 and VGLUT3 in undifferentiated or differentiated MPC homogenates (0.3 µg mouse brain membrane, 30 µg undifferentiated or differentiated MPCs’ homogenate for VGLUT1 and VGLUT3 per lane). (**C**) Immunofluorescent staining of VGLUT1 and VGLUT3 in undifferentiated MPCs. The boxed areas in the overlay images are enlarged. Bar: 10 µm. (**D**) Immunofluorescent staining of VGLUT1 and VGLUT3 in differentiated MPCs. The boxed areas in the overlay images are enlarged. Bar: 20 µm. (**E**) Colocalization rates of VGLUT3 with VGLUT 1 in undifferentiated and differentiated MPCs. Data are represented as the mean ± S.E.M. of more than 30 cells in three independent experiments. For each sample, colocalization was determined in three randomly selected areas per cell (25 µm^2^).

**Figure 3 ijms-26-02485-f003:**
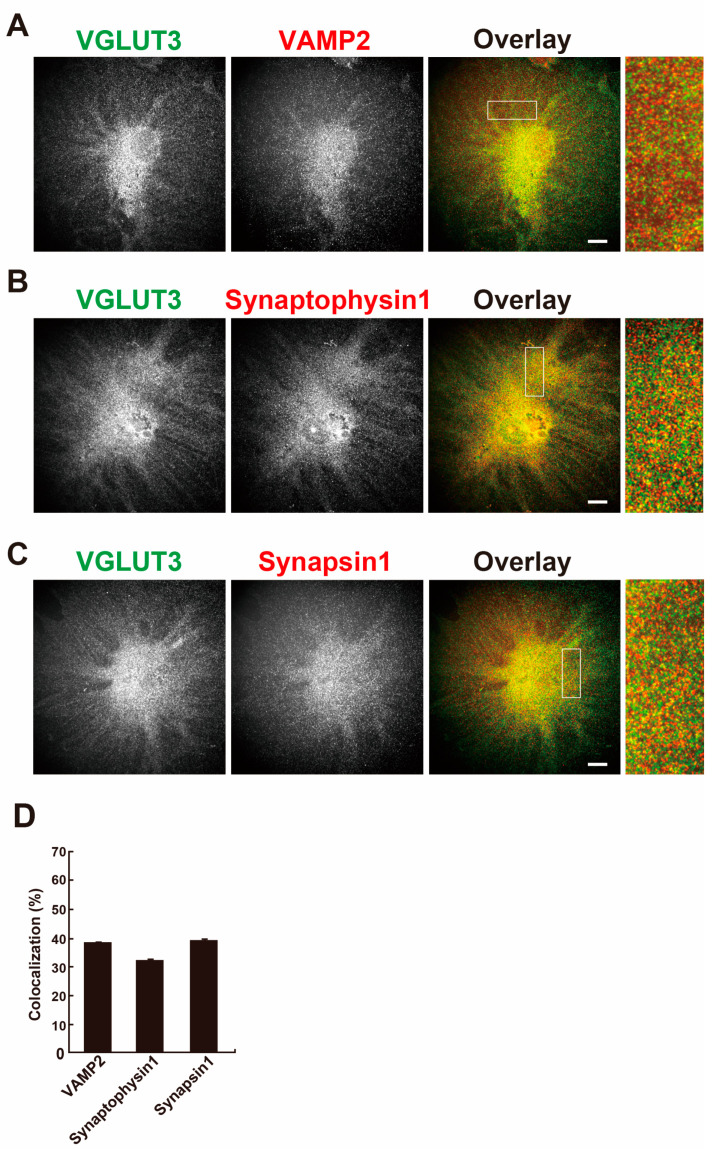
Synaptic vesicle-associated proteins are colocalized with VGLUT3 in differentiated MPCs. The synaptic vesicle-associated proteins VAMP2 (**A**), synaptophysin 1 (**B**), and synapsin 1 (**C**) colocalized with VGLUT3. VGLUT3, VAMP2, synaptophysin 1, and synapsin 1 were visualised by double immunofluorescence. The boxed areas in the overlay images are enlarged. Bar: 20 µm. (**D**) Colocalization rates of VGLUT3 with synaptic vesicle-associated proteins in undifferentiated and differentiated MPCs. Data are represented as the mean ± S.E.M. of more than 30 cells in three independent experiments.

**Figure 4 ijms-26-02485-f004:**
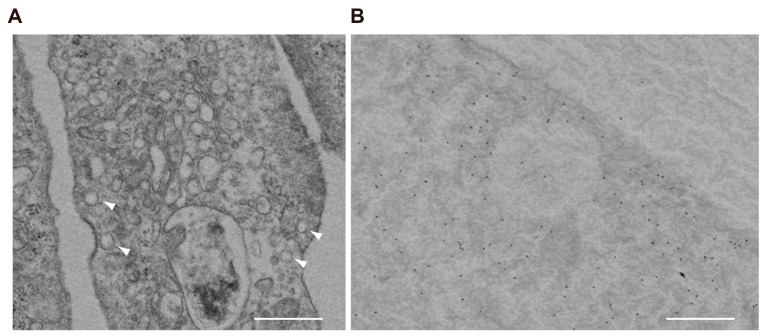
VGLUT3 is localized at the membrane of vesicle-like structures in differentiated MPCs. (**A**) Electron micrograph of differentiated MPCs fixed with osmium tetroxides. The arrowheads indicate vesicle-like structures. Scale bar: 500 nm. (**B**) Immunogold electron micrograph of differentiated MPCs stained with anti-VGLUT3 antibodies. Scale bar: 500 nm.

**Figure 5 ijms-26-02485-f005:**
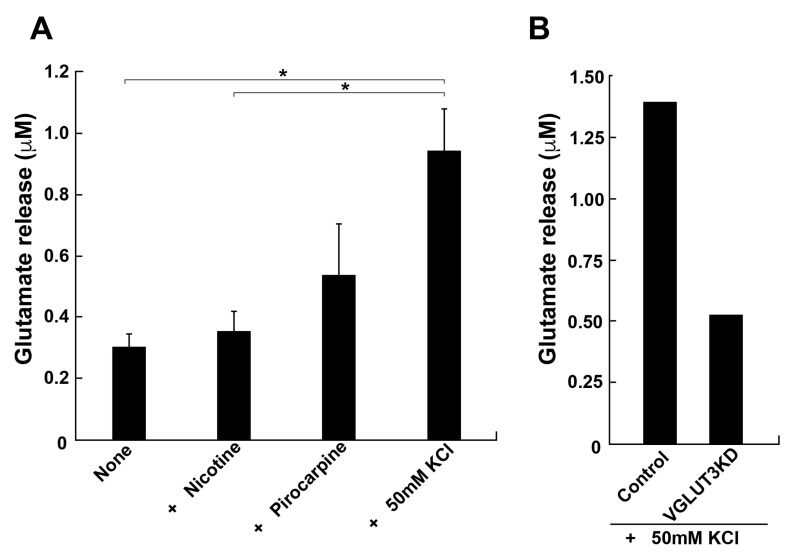
Glutamate release by differentiated MPCs is partially VGLUT3-dependent. (**A**) Glutamate release by differentiated MPCs. Differentiated MPCs in Ringer’s solution were stimulated with 100 µM nicotine, pilocarpine, or high K^+^ Ringer’s solution (50 mM KCl) at 37 °C for 5 min, and then released glutamate was measured by HPLC. Data are represented as the mean ± S.E.M. of three independent experiments. *: *p* < 0.05. (**B**) Depletion of VGLUT3 by RNAi reduces KCl-induced glutamate release. Control and VGLUT3-knockdown differentiated MPCs in Ringer’s solution were stimulated with 50 mM KCl, as described in A. Released glutamate release was measured by HPLC. Data are represented as the average of two independent experiments.

## Data Availability

The raw data supporting the conclusions of this article will be made available by the authors without undue reservation.

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
