# Peer review of "Vesicular Glutamate Transporter 3 Is Involved in Glutamatergic Signalling in Podocytes"

_ijms, 2025, doi:10.3390/ijms26062485_

Round 1

Reviewer 1 Report

Comments and Suggestions for Authors

The authors  describing in their manuscript the vesicular glutamate transporter 3 (VGLUT3)-dependent glutamate release from podocytes. Immunofluorescence analysis revealed that rat glomerular podocytes and an immortal mouse podocyte cell line (MPCs) express VGLUT1 and VGLUT3. Consistent with this finding, quantitative RT-PCR revealed the expression of VGLUT1 and VGLUT3 mRNA in undifferentiated and differentiated MPCs. In addition, the exocytotic proteins vesicle-associated membrane protein 2, synapsin 1, and synaptophysin 1 were present in punctate patterns, and colocalised with VGLUT3 in MPCs.Differentiated MPCs released glutamate following depolarization with high potassium levels and after stimulation with the muscarinic agonist pilocarpine. Depletion of VGLUT3 in MPCs by RNA interference reduced depolarization-dependent glutamate release. These results strongly suggest that VGLUT3 is involved in glutamatergic signalling in podocytes and maybe a new drug target for various kidney diseases.

Overall, the manuscript is very well written and provides an actual overview of the current understanding of the role of the vesicular glutamate transporter functions in glutamatergic signalling in podocytes. Their experimental findings give a deeper understanding that VGLUT3 is involved in glutamatergic signalling in podocytes, which may open new doors for treatment options for kidney diseases.

Introduction: The authors introduce to the topic summarizing the essentials aspects to demonstrate the importance of the glutamatergic system in podocytes: kidney contains cells, including podocytes, which transmit glutamate signals. Overall, a detailed overview of the current knowledge of the NMDA/Glutamate system with focus on the glutamate transporters is provided. However, little is known about its molecular mechanisms. After the general introduction to the transporter system the main focus of the current study is provided;

In this study, the authors investigated the functional expression of VGLUT3 in differentiated mouse podocytes (MPCs) and observed that VGLUT3 largely colocalised with VGLUT1 and VAMP2, as well as the synaptic vesicle marker proteins synaptophysin 1 and synapsin 1. Furthermore, immuno-electron microscopy showed that VGLUT3 was present on the membrane of vesicle-like structures. Differentiated MPCs secreted glutamate into extracellular space upon the stimulation with pilocarpine or by depolarization with KCl. Depletion of VGLUT3 in differentiated MPCs decreased the KCl-induced glutamate release.

The result section is well structured and is divided in 5 sub-sections:

  • VGLUT3 is present in glomerular podocytes
  • VGLUT1 and VGLUT3 are co-expressed in differentiated MPCs
  • VGLUT3 colocalises with synaptic vesicle marker proteins in differentiated MPCs
  • VGLUT3 associates with synaptic-like vesicles in differentiated MPCs
  • VGLUT3 depletion in differentiated MPCs reduces glutamate release into the intracellular space

Several figures are shown to support the understanding of the findings including 2 additional ones as supplementary files. All figures are well selected and all relevant information’s is provided in the figure legends. The expression of VGLUT1 and VGLUT3 in MPCs as well as renal glomerular podocytes is well documented. Overall, the result section is clearly structured and well described. All findings are presented in an excellent way.

Discussion: The results and the outcome of the work is discussed with respect to the historical findings of the VGLUT isoforms and their physiological role. References to previous work is well selected, however, some more references may be useful to consider also other historical and current findings in this field. This section lacks a little bit of clarity. It is not easy to see what is known and what are the current findings. This section may need a revision to get a better differentiation of impact of the current findings within the current understanding. Correctly the authors are closing the discussion section with the comment that further studies may be needed to get a better understanding of stimuli cause glutamate release from podocytes. The fact that podocytes release glutamate via the activity of VGLUT3 is the findings of the current investigations. At least some thoughts about potential mechanisms of the release mechanisms would enhance the importance of the paper. In addition, some more information should be provided in what direction a potential drug target can contribute to novel treatment options.

Materials and Methods section: All relevant parts are addressed in the respective 11 sub-sections:

  • Antibodies and reagents
  • Cell culture
  • siRNA-mediated interference and transfection
  • Quantitative PCR
  • Immunohistochemistry
  • Electron microscopy
  • Fluorescent microscopy
  • Determination of glutamate release
  • Morphometry
  • Ethics and animal use statement
  • Statistical analysis

All details are provided. The section is comprehensive and complete. No further comment!

Conclusion: The conclusion is very short and a repetition of the abstract and the discussion section: “Podocytes themselves release glutamate via the activity of VGLUT3.”. A classification in the current state of knowledge as well as an outlook for further studies may improve the overall quality of the manuscript, irrespectively of the experimental findings, which are very well documented.

Author Response

Reviewer #1

Comment 1;

The manuscript describes the possible role of the vesicular glutamate transporter 3 (VGLUT3) in glutamatergic signaling in podocytes. The authors have delved into the expression of VGLUTl and VGLUT3 mRNA in undifferentiated and differentiated MPCs through quantitative PCR, Western blot analysis, and immunostaining. Furthermore, its was shown that glutamate signaling and the signal reception machinery expressed by mature podocytes are crucial for controlling blood filtration function, and VGLUT3 is an essential component of this machinery. Overall, it was found that podocytes themselves release glutamate via the activity of VGLUT3, thus attributing a specific role for that molecule. The work was carried out competently by the authors and deserves further consideration. One striking observation pertains to Figure 2B. What is shown there is actually the Western blot analyses of VGLUTl and VGLUT3 in undifferentiated or differentiated MPC homogenates. It appears that the band observed for VGLUT3 for the differentiated MPCs is weaker than that of the undifferentiated MPCs. That is in contrast to the corresponding behavior of the VGLUT3 vs that of the undifferentiated counterpart in the abutting diagram on the quantitation of the mRNA (Figure 2A). That is not clear and should be justified.

Based on the remarks made, the point raised should be justified and appropriate revision should

ensue. Then, the manuscript could be considered.

Response;

Thank you for the valuable comment.  It is known that activity of many proteins including synaptopodin, CD2AP and dynamin is regulated by degradation with several proteases in mature podocytes (Reviewed by Rinschen et al., The podocyte protease web: uncovering the gatekeepers of glomerular disease. Am J Physiol Renal Physiol. 315, F1812-F1816, 2018). It is likely that the VGLUT3 protein is also degraded, and its amount is regulated. We added the description in “Discussion” (Page 9. line282-287) and one reference.

Reviewer 2 Report

Comments and Suggestions for Authors

Review comments are attached

Author Response

Reviewer #2

Comment 1;

Discussion: The results and the outcome of the work is discussed with respect to the historical findings of the VGLUT isoforms and their physiological role. References to previous work is well selected, however, some more references may be useful to consider also other historical and current findings in this field. This section lacks a little bit of clarity. It is not easy to see what is known and what are the current findings. This section may need a revision to get a better differentiation of impact of the current findings within the current understanding.

Response;

Thank you for the comment. According to the referee’s advice, we added the related descriptions about glutamatergic signal transmission in podocyte in the “Discussion” (Page 9, line 271- 278).

Comment 2;

Correctly the authors are closing the discussion section with the comment that further studies may be needed to get a better understanding of stimuli cause glutamate release from podocytes. The fact that podocytes release glutamate via the activity of VGLUT3 is the findings of the current investigations. At least some thoughts about potential mechanisms of the release mechanisms would enhance the importance of the paper.

Response;

According to the referee’s comment, we discussed potential release mechanism and added related descriptions in the “Discussion” (Page 9, line 277; Page 9, line 288- Page 10, line 297).

Comment 3;

In addition, some more information should be provided in what direction a potential drug target can contribute to novel treatment options.

Response;

Thank you for the valuable comment. Possible drug targets were discussed in the “Discussion” (Page 10, line 320- 323).

Comment 3;

Conclusion: The conclusion is very short and a repetition of the abstract and the discussion section: “Podocytes themselves release glutamate via the activity of VGLUT3.”. A classification in the current state of knowledge as well as an outlook for further studies may improve the overall quality of the manuscript, irrespectively of the experimental findings, which are very well documented.

Response;

Thank you for the comment. We rewrote to include all the right information. (Page 15, line 483- 488).